# Selective excitation of vibrations in a single molecule

Yang Luo [1] ✉, Shaoxiang Sheng [1], Michele Pisarra[2,3], Alberto Martin-Jimenez[1,4], Fernando Martin [4,5] ✉, Klaus Kern [1,6] & Manish Garg [1] ✉

The capability to excite, probe, and manipulate vibrational modes is essential for understanding and controlling chemical reactions at the molecular level. Recent advancements in tip-enhanced Raman spectroscopies have enabled the probing of vibrational fingerprints in a single molecule with Ångström-scale spatial resolution. However, achieving controllable excitation of specific vibrational modes in individual molecules remains challenging. Here, we demonstrate the selective excitation and probing of vibrational modes in single deprotonated phthalocyanine molecules utilizing resonance Raman spectroscopy in a scanning tunneling microscope. Selective excitation is achieved by finely tuning the excitation wavelength of the laser to be resonant with the vibronic transitions between the molecular ground electronic state and the vibrational levels in the excited electronic state, resulting in the state-selective enhancement of the resonance Raman signal. Our approach contributes to setting the stage for steering chemical transformations in molecules on surfaces by selective excitation of molecular vibrations.

The vibrational modes of a molecule play a crucial role in its chemical (and geometrical) transformation[1,2] as well as in many energy conversion phenomena[3–5]. Various theoretical and experimental studies have demonstrated the pivotal role of selectively exciting molecular vibrations in achieving precise control over the desired output of chemical transformations and electron transport processes[6–8]. To comprehend the elementary mechanisms of molecular reactions, much effort has been directed toward exciting vibrational modes and modifying chemical reactions at the single-molecule level[9–11]. Scanning tunneling microscopy (STM) has emerged as a powerful technique to investigate single-molecule events related to specific vibrational modes in real-space, utilizing inelastic electron tunneling scattering[12–14]. Nevertheless, the broad energy distribution of tunneling electrons will inevitably populate the excited states of the molecule within a wide energy range, thus making it difficult to unequivocally probe one specific vibrational mode.

The selective excitation of vibrational modes in molecules is usually achieved by resonant infrared laser radiation[15,16] or by stimulated Raman pumping[17,18]. However, the combination of infrared spectroscopy with scanning probe microscopy can hardly achieve sub-nanometer resolution[19], which is crucial for single-molecule studies. The recent advancements in integrating STM with visible and near-infrared light have expanded the capability of manipulating vibrational degrees of freedom at the single-molecule level[20–23]. STM based tip-enhanced Raman spectroscopy (TERS)[24–28] has demonstrated Ångström-scale resolution in probing the spatial distribution of vibrational modes in a single molecule. Nevertheless, this enhanced Raman spectroscopy involves inelastic scattering from numerous vibration modes instead of specific (selective) modes.

A promising approach to enhance particular molecular vibrations is through resonance Raman scattering, which relies on the coupling

[1]Max Planck Institute for Solid State Research, Heisenbergstr. 1, 70569 Stuttgart, Germany. [2]Dipartimento di Fisica, Università della Calabria, Via P. Bucci, Cubo 30C, 87036 Rende, CS, Italy. [3]INFN-LNF, Gruppo Collegato di Cosenza, Via P. Bucci, Cubo 31C, 87036 Rende, CS, Italy. [4]Instituto Madrileño de Estudios Avanzados en Nanociencia (IMDEA Nano), Faraday 9, Cantoblanco, 28049 Madrid, Spain. [5]Departamento de Química, Módulo 13, Universidad Autónoma de Madrid, 28049 Madrid, Spain. [6]Institut de Physique, Ecole Polytechnique Fédérale de Lausanne, 1015 Lausanne, Switzerland. ✉e-mail: y.luo@fkf.mpg.de; fernando.martin@imdea.org; mgarg@fkf.mpg.de

between vibrational and electronic states in molecules[29–34]. By tuning the excitation wavelength to fall within the optical absorption band of the molecules, specific vibrational modes coupled to an excited electronic state can be selectively excited. This technique has been successfully applied to reveal structural changes and functional dynamics of photoactive chromophores[35–37]. Moreover, by aligning the excitation photon energy to different vibronic transitions, resonant Raman scattering can be used to precisely determine the electronic and vibrational structure of molecules in the bulk phase[38,39], and to study the exciton-phonon interactions in low-dimensional materials[40–42].

In this work, we employ tip-enhanced resonance Raman scattering to investigate the mode selective excitation in a single molecule. Resonance Raman scattering and fluorescence signals were simultaneously measured from deprotonated phthalocyanine molecules (HPc$^-$) adsorbed on a thin insulating film on top of a metallic substrate. The contributions of the Raman scattering and fluorescence processes are identified by their emission peak positions and linewidths in the recorded spectra. We demonstrate selective excitation of the vibrational modes by tuning the excitation wavelength of the laser to be resonant with the corresponding molecular vibronic transitions between the ground electronic state and specific vibrational modes in the excited electronic state. The mode selective excitation manifests as a strong enhancement of particular vibrational modes (associated with the chosen vibronic transitions) in the recorded resonance Raman spectra. To elucidate the physical mechanism underlying the mode selective excitation, we performed first-principles density functional theory (DFT) based simulations accounting for the Raman transitions involving different modes in the excited and ground electronic states of the molecule, which agree well with the experimental findings. Furthermore, we performed spatial dependence measurements of the resonance Raman scattering to illustrate the Ångström-scale resolution in selective excitation of molecular vibrations.

## Results

### Single-molecule spectroscopy

In our experiments, we used STM-based near-field spectroscopic techniques including tip-enhanced photoluminescence (TEPL)[43–47] and STM-induced electroluminescence (STML)[48–52] to characterize the optical spectral properties of single free-base phthalocyanine (H$_2$Pc) molecules and their deprotonated species (HPc$^-$), as illustrated in Fig. 1a. A gold tip and an Au(111) substrate were used to achieve a strong plasmonic enhancement for the used laser wavelength (see "Methods"). An ultrathin three-monolayer (3 ML) sodium chloride (NaCl) insulating film was deposited on the Au(111) surface to electronically decouple the single molecules from the underlying Au substrate, which efficiently suppresses the quenching rates of the excited molecular excitons.

Figure 1b shows the STML spectrum of an H$_2$Pc molecule (blue curve in the top panel of Fig. 1b) locally excited by tunneling electrons at the bias voltage of 2.4 V. A main emission peak at ~1.815 eV with a full width at half maximum (FWHM) of ~8 meV was measured, which is assigned to a purely electronic contribution, usually referred to as Q$_x$, in the phthalocyanine molecules[53–56] (Supplementary Fig. 7). The molecular TEPL spectrum was measured by using a continuous wave (CW) laser centered at ~633 nm at a bias voltage of −1.0 V, as shown by the red curve in the top panel of Fig. 1b. Here, the energy of tunneling electrons (up to 1 eV) is much lower compared to the electronic transition energy (~1.8 eV) of the H$_2$Pc molecule, which is not sufficient to excite the H$_2$Pc molecule directly as in the STML measurement[57]. In contrast to the STML spectrum showing a single broad fluorescence peak, the TEPL spectrum of the H$_2$Pc molecule shows additional distinct narrow peaks over a broad peak.

A deprotonated version of the H$_2$Pc molecule was also investigated by STML and TEPL measurements[54,58]. The deprotonation of an H$_2$Pc molecule is achieved by positioning the STM tip on top of the

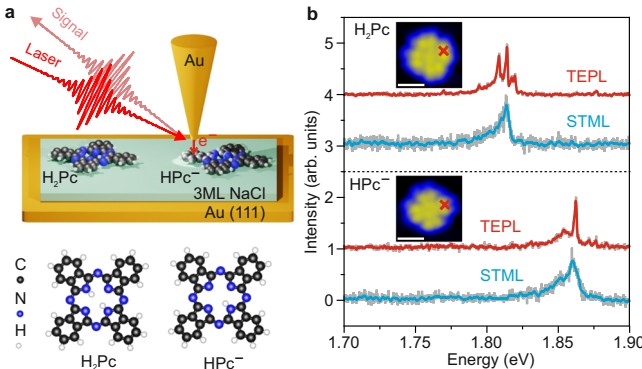

**Fig. 1 | Single-molecule spectroscopy. a** Top panel: schematic illustration of the experimental setup for STM-induced electroluminescence (STML) and tip-enhanced photoluminescence (TEPL) measured from single molecules adsorbed on three-monolayer thick NaCl on top of Au(111). Bottom panel: the structures of the H$_2$Pc and HPc$^-$ molecules. **b** STML (blue curves) and TEPL (red curves) spectra of individual H$_2$Pc and HPc$^-$ molecules. The STML spectrum of H$_2$Pc was recorded with tunneling electron excitation at a bias voltage $V = 2.4$ V, a tunneling current $I = 100$ pA with an acquisition time $t = 5$ s. The STML spectrum of HPc$^-$ is measured at $V = 2.4$ V, $I = 20$ pA, and $t = 60$ s. The TEPL spectra were recorded with CW laser excitation ($\lambda$ - 633 nm and laser power of 0.2 mW) at $V = -1$ V, $I = 4$ pA, and $t = 10$ s. All spectra are normalized and vertically shifted for clarity. The insets show the measured STM topography images of the H$_2$Pc and HPc$^-$ molecules adsorbed on the three-monolayer thick NaCl/Au(111) surface, acquired at $V = -2$ V and $I = 4$ pA. The STML and TEPL spectra were recorded by placing the nanotip above the molecular lobe indicated by the red crosses in the topographic images. The white scale bars indicate a length of 1 nm.

molecular center and applying a bias voltage of ~3 V. The STML spectrum of the HPc$^-$ molecule shows a blue-shifted fluorescence peak at ~1.86 eV (blue curve in the bottom panel of Fig. 1b) compared to the H$_2$Pc molecules, plausibly induced by the internal Stark effect owing to the electrostatic field generated by the internal charges of the deprotonated HPc$^-$ molecule[54]. The TEPL spectrum of the HPc$^-$ molecule shows sharp peaks sitting on top of a broad peak, similar to the spectral features measured from the H$_2$Pc molecule.

The broad peaks in the TEPL spectra for both types of molecules can be attributed to the fluorescence emission, whose linewidth is determined by the total electronic damping time, which is usually ~10 meV for single molecules in the STM junction[43]. The appearance of narrow emission peaks overlaying on top of the broad fluorescence background (Fig. 1b) implies the involvement of either the Raman scattering processes or the vibronic transitions between the vibrational levels of the involved electronic states, i.e., excited and ground electronic states, in the TEPL spectra.

### Selective excitation of molecular vibrations

To decipher the origin of the narrowband emission lines appearing in the TEPL spectra, we tuned the excitation wavelength of the laser in the TEPL measurements. While tuning the laser's wavelength will not alter the vibronic transition energies as they are intrinsic molecular properties, it will shift the energies of Raman scattering light, which depend on the energy differences of the incident laser photon energy and the molecular vibrational modes[59]. For this experiment, we excited the molecule using a continuously wavelength-tunable picosecond fiber laser (see "Methods" for details). We note that the H$_2$Pc molecule is prone to deprotonation upon photoexcitation with the picosecond laser, possibly caused by the strong photo-induced electron tunneling on interaction with the ultrashort laser pulses. Thus, we focused on the HPc$^-$ molecule in the wavelength-dependent experiments.

The TEPL spectra of an HPc$^-$ molecule were recorded at various excitation wavelengths ranging from 631 nm to 641.5 nm as shown in

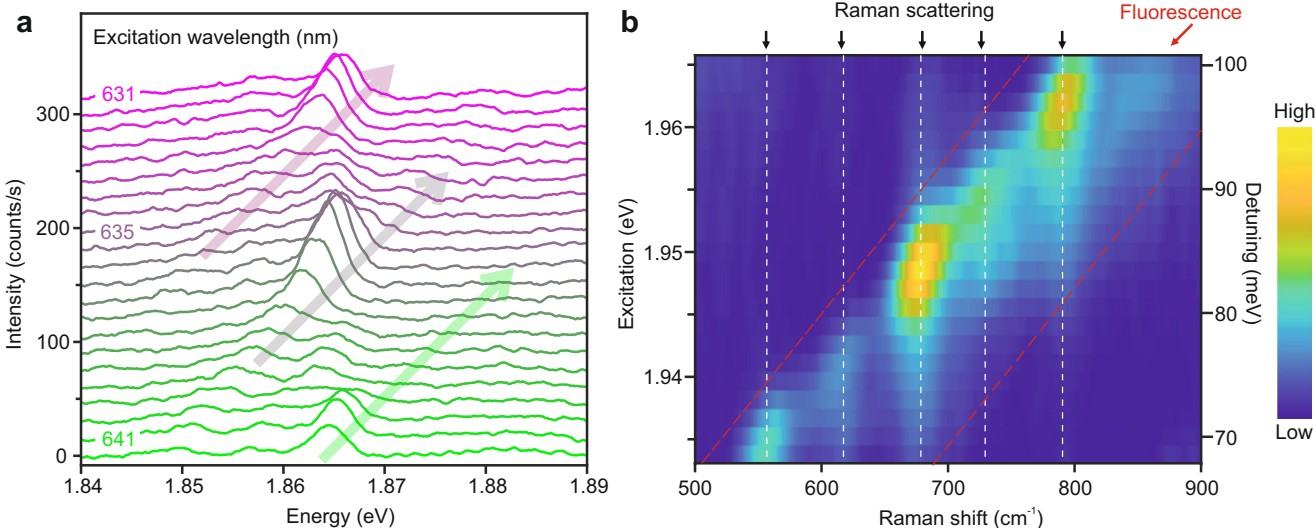

**Fig. 2 | Wavelength-dependent photoexcitation of a single HPc⁻ molecule. a** A series of TEPL spectra of a single HPc⁻ molecule recorded by tuning the excitation wavelength of the laser. The laser wavelength was tuned from 631 nm to 641.5 nm (indicated by purple to green colors) with a step size of 0.5 nm, and the laser power was kept constant at 0.23 mW. The TEPL spectra were measured by placing the nanotip above the molecular lobe (red cross in Fig. 1b) at $V = -1.5$ V, $I = 4$ pA, and $t = 20$ s. The three arrows indicate the spectral shifting of the narrow emission peaks upon changing the excitation wavelength of the laser. The spectra are vertically shifted for clarity. **b** Raman excitation map generated from the recorded TEPL spectra shown in (**a**). The x-axis is the Raman shift frequency (cm⁻¹), and the left y-axis is the tunable laser excitation photon energy (eV). The right y-axis denotes the energy detuning (in meV) between the incident laser photon energy and the TEPL peak at -1.865 eV. The black arrows on top of the figure and the vertical dashed-white lines indicate the spectral positions of the Raman peaks, whereas the red arrow and the space between the tilted dashed-red lines indicate the position of the broad fluorescence emission.

Fig. 2a. We note that the picosecond laser pulses have a spectral line-width of ~1.5 nm (~5 meV at 640 nm), which reduces the spectral resolution in the TEPL spectra as compared with the CW laser excitation (Fig. 1b). As a result, the bandwidth of the narrow emission peaks in Fig. 2a is limited to ~5 meV. An apparent gradual shift of the narrow emission peaks, as indicated by the colored arrows in Fig. 2a, can be clearly seen upon decreasing the excitation wavelength of the laser. As explained below, this indicates that the measured narrow emission peaks overlapping with the fluorescence peak have their origin in Raman scattering processes.

By converting the horizontal axis in Fig. 2a from photon energy (eV) to the relative Raman shift frequency (cm⁻¹) with respect to the laser excitation wavelength, we obtain the wavelength-dependent Raman excitation map, as shown in Fig. 2b. The broad fluorescence peak, which is spectrally invariable in the photon energy representation, appears to gradually shift with the excitation wavelength in the Raman shift representation. Notably, several narrow peaks at around 555, 615, 680, 725, and 790 cm⁻¹ (indicated by black arrows), with fixed Raman shift energies appearing on top of the broad fluorescence peak can be clearly identified. This constant energy shift with excitation wavelength clearly shows that the narrow emission peaks overlapping with the fluorescence emission have their origin in the Raman scattering processes. A striking feature in the Raman excitation map is that the observed Raman peaks undergo an intensity modulation upon the variation of the excitation wavelength of the laser.

To understand the underlying mechanism of the observed features in the experiment, we illustrate in Fig. 3a the energy diagrams of the fluorescence and Raman scattering processes (see Supplementary Section I and Supplementary Fig. 1 for details). In the fluorescence process, the molecule is photo-excited from its ground electronic state $|0, g\rangle$ to the vibrational levels in the excited electronic state $|\nu_e, e\rangle$. Following photoexcitation, the molecule can redistribute the excess vibrational energy among its various modes through an internal conversion process or exchange energy with its environment to reach the bottom of the excited electronic state, i.e., $|0, e\rangle$, whence fluorescence emission to the ground electronic state takes place. On the other hand,

the Raman scattering process connects the initial $|0, g\rangle$ state and the final, $|\nu_g, g\rangle$ states, without involving a direct photon absorption in the molecule. Yet, the Raman signal can be dramatically enhanced when the excitation wavelength of the laser is resonant with the electronic absorption band, a process referred to as resonance Raman scattering[29,33].

Within the Born-Oppenheimer approximation, the resonant Raman scattering cross-section is mainly determined by two vibronic transitions: (1) from $|0, g\rangle$ to $|\nu_e, e\rangle$, and (2) from $|\nu_e, e\rangle$ to $|\nu_g, g\rangle$. When the energy of the incident photons matches a vibronic transition, e. g., $|0, g\rangle$ to $|\nu_e, e\rangle$, the transition rate is maximized, leading to an increase in the Raman scattering cross-section. For instance, the resonant excitation of the $\nu_I$ vibrational mode leads to a strong resonant Raman signal from $\nu_I$, as shown in Fig. 3b (left panel), while resonant excitation of the $\nu_{II}$ vibrational mode at higher photon energy leads to an enhancement of Raman scattering from $\nu_{II}$ (right panel). In this way, the selective enhancement of Raman scattering from a particular vibrational mode can be achieved by tuning the laser photon energy to be resonant with its related vibronic transition. The intensity of the fluorescence and Raman lines are further modulated by the Franck-Condon (FC) overlap integrals involving the vibrational wave functions of the upper and lower electronic states, $\langle \nu_g | \nu_e \rangle$. When the geometries of the molecule in the ground and excited electronic states are similar, the energy spacing of the vibrational levels in both electronic states are also very similar and the largest Franck-Condon overlaps correspond to transitions involving identical vibrational modes in the two electronic states (indicated by thick arrows in Fig. 3a, b). As a consequence, the energy difference of the dominant Raman transition will be nearly identical to that of the fluorescence transition, which is what we observe in our experiment (Fig. 2a).

Following the concept of resonance Raman scattering, we performed time-dependent DFT-based first-principles simulations (see Supplementary Section II for details) considering the Raman scattering between the vibronic levels of the ground and excited electronic states of the HPc⁻ molecule. For comparison with the experiment, the excited singlet electronic state with the lowest transition energy (S₁),

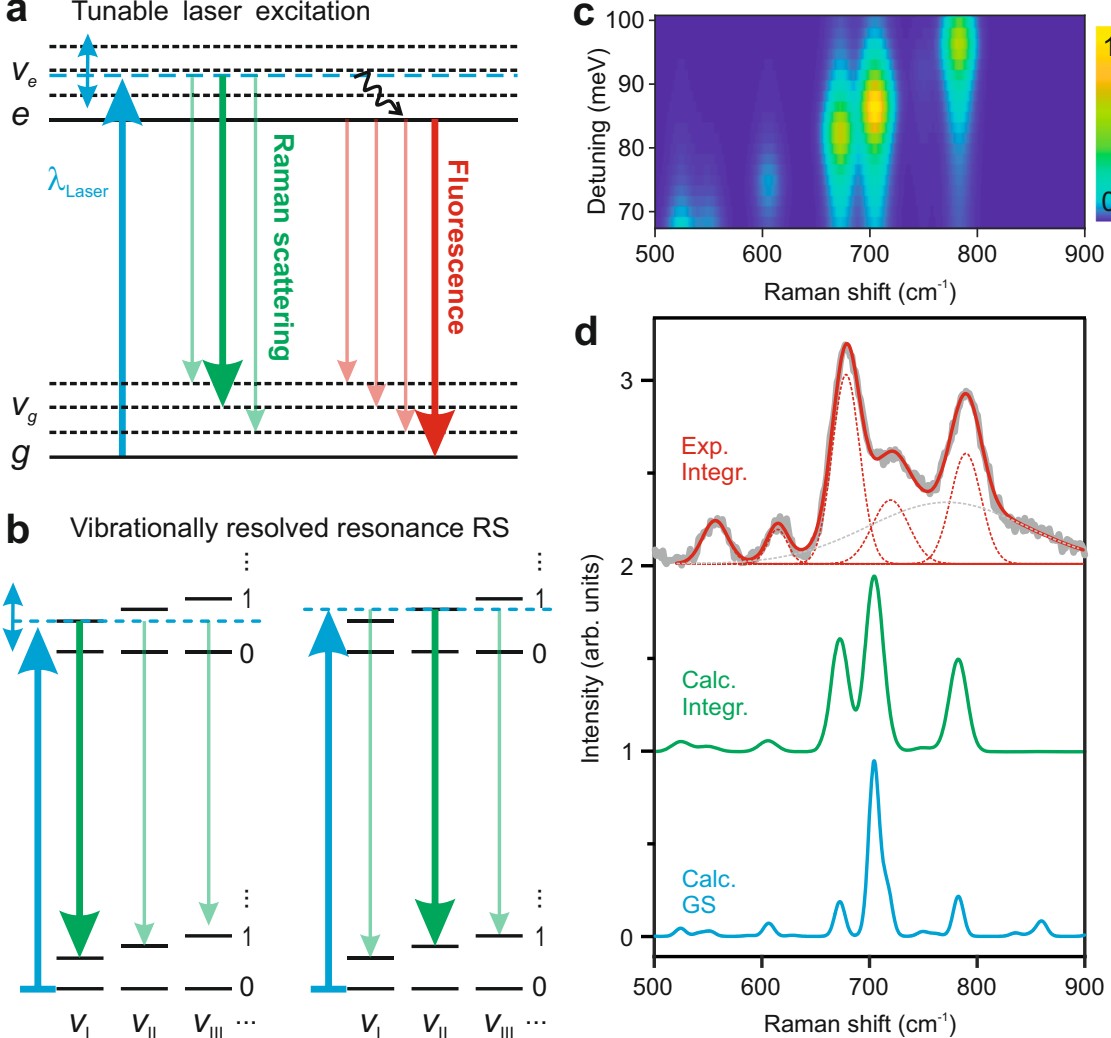

**Fig. 3 | Vibrationally resolved resonance Raman spectroscopy of a single molecule. a** Schematic illustration of the resonance Raman scattering and fluorescence in a molecule with tunable laser excitation (blue upward arrow). The blue double arrow indicates the tunable energy range, and the black wave arrow indicates the vibrational relaxation. The Raman transitions are represented by green curves, whereas the fluorescence processes are indicated by the red curves. The relative intensities of different transitions are symbolized by the thickness of the downward arrows. **b** Schematic illustration of the selective enhancement of Raman scattering (RS) from particular vibrational mode, $v_I$ or $v_{II}$, achieved by tuning the laser photon energy to be resonant with its related vibronic transition. Numerals, 0, 1... represent the quantum numbers of the various vibrational modes (e.g., $v_I$, $v_{II}$ or

$v_{III}$) in the two electronic states. The molecule is initially in the vibrational ground state before the laser excitation. **c** Simulated Raman excitation map generated by calculating the resonance Raman scattering spectra at various energy detunings between the incident photons and the calculated optical gap between the lowest vibrational states of the two electronic states $g$ and $e$. **d** Comparison between the experimental (black curve) and simulated (green curve) integrated Raman spectra, obtained by summing up all the TEPL spectra in the Raman shift representation acquired at different excitation wavelengths. The blue curve shows the calculated Raman spectrum for the ground electronic state. The spectra are vertically shifted for clarity.

corresponding to the $Q_x$ state as measured experimentally, was considered. The calculations include explicitly the dipole couplings between the ground ($g$) and the excited ($e$) electronic states and the Franck-Condon overlap integrals, $\langle v_g | v_e \rangle$. The resonant Raman scattering spectra were obtained by calculating the Raman scattering cross-sections at various excitation energies. Figure 3c shows the simulated Raman spectrum as a function of the energy detuning between the incident photons and the calculated optical gap between the lowest vibronic states $|0, g\rangle$ and $|0, e\rangle$. The simulated result matches quite well with the experimental Raman excitation map (Fig. 2b), which demonstrates the selective excitation of vibrational modes using resonant Raman scattering processes (Supplementary Fig. 5). This achievement stems from the precise control of the incident photon energy to align with a particular molecular vibronic transition

rather than the $|0, g\rangle$ to $|0, e\rangle$ one. We note that non-selective resonant excitation of Raman modes in a single molecule has been earlier demonstrated by resonant excitation of this $|0, g\rangle$ to $|0, e\rangle$ transition, where all the Raman modes coupled to the excited state were enhanced[21]. It is worth emphasizing that the dephasing time of the vibronic states is sufficiently long in the experiments, which enables the spectral separation of different vibronic transitions and the observation of the dominance of specific Raman transitions from a single vibrational mode.

To obtain the energies of the molecular vibrational modes involved in the resonance Raman scattering processes, we summed up the TEPL spectra in the Raman excitation maps (Figs. 2b and 3c) to produce the integrated Raman spectrum of the HPc⁻ molecule, as shown in Fig. 3d. The relative intensities of different vibrational modes

in the generated spectrum is a consequence of the varying strengths of the Frank-Condon overlap integrals for different vibronic transitions over the tuned excitation wavelength. Here, we focus on the vibrational modes between 500 and 900 cm$^{-1}$ and the associated molecular vibronic transitions. Five distinct Raman modes can be distinguished by fitting the experimental data (black curve) with five narrow Gaussian peaks (dashed-red curves) on a broad Gaussian background (dashed-gray curve). The experimentally measured vibrational features can be well-reproduced in the simulated Raman spectra (green curve) obtained by summing up all the TEPL spectra in the simulated Raman excitation map (Fig. 3c), especially for the frequencies of the vibrational modes. The same Raman modes are observed in the calculated non-resonant Raman spectrum for the ground electronic state, as shown by the blue curve in Fig. 3d. In the experiment, the molecules are subjected to a highly localized plasmonic field in the STM junction, while in the DFT calculations the field is assumed to be spatially uniform and with no spectral distribution. Both spatial and spectral distributions of the field[26,27] and the local geometry of the molecule[60] can affect the measured spectral features, which may contribute to the differences between the relative spectral intensities of the measured and calculated spectra.

### Ångström-scale resolution in selective excitation of molecular vibrations

To explore the extent of spatial confinement of selective excitation of molecular vibrations in the STM junction, TEPL spectra of a single HPc$^-$ molecule were recorded by increasing the plasmonic gap size, and by moving the nanotip laterally over the molecule (Fig. 4a, b). The molecule was excited by picosecond laser pulses set at ~633.5 nm. Figure 4c shows a series of spectra measured by increasing the plasmonic gap size, i.e., by reducing the tunneling current under the constant current operating mode of the STM. The sharp peak at ~1.86 eV originates from the resonant Raman scattering for the vibrational mode present at ~790 cm$^{-1}$, and the broad peak results from the fluorescence emission. The intensities of resonance Raman scattering (black squares) and fluorescence (red dots) contributions in the measured spectra were obtained by integrating the spectral intensities over the area under the Raman and fluorescence peak positions, both of which decay exponentially with increasing plasmonic gap size, as shown in Fig. 4d. Fitting the intensity profiles with an exponential function (solid curves), $P \propto \exp(-\Delta z/k)$, yields decay constants of $k$ ~ 0.25 nm for both of the contributions in the measured spectra, suggesting a very similar dependence of the resonance Raman and fluorescence signals on the plasmonic gap size. This is because the Raman scattering and fluorescence rates have the same dependence on the electronic transition dipolar moments between the ground and the excited electronic states. Such similarity arises from the same dependence of the electronic transition dipolar moments between $g$ and $e$ on both Raman scattering and fluorescence rates[30].

A series of TEPL spectra were recorded by positioning the nanotip along the blue crosses over the molecule, as shown in Fig. 4e. The integrated resonance Raman scattering (black squares) and fluorescence intensities (red dots) are plotted as a function of the lateral distance of the nanotip from the molecular center, as shown in Fig. 4f. The spatial confinement of the fluorescence emission shows a minimum at the molecular center, while it is maximum at the edge of the molecule. This emission behavior is determined by the dipolar interaction between the nanocavity plasmons and the molecule[43]. The integrated spectral intensities in Fig. 4e, f are plotted in absolute terms, implying that no TEPL signal is measured at the center of the molecule, pointing out the importance of achieving spatial resolution for an optimum vibrational selectivity. The spatial variation of the Raman signal shows similar feature with that of the fluorescence signal. From the evolution of the Raman scattering intensity change

at various nanotip positions, we can estimate a spatial resolution of ~5 Ångström in the selective excitation of molecular vibrations in a single molecule.

## Discussion

In conclusion, we have demonstrated selective excitation of vibrations in a single molecule utilizing STM-based tip-enhanced resonance Raman scattering. By tuning the excitation wavelengths of the incident laser to resonance with the corresponding vibronic levels in the excited state, we can selectively excite and probe a specific vibrational mode in a single molecule with Ångström-scale resolution. This methodology provides a unique avenue for probing the photoexcitation processes in single molecules, offering insights into the intricate interactions between light and matter at the sub-molecular level. Moreover, our ability to control the population of selective vibrational modes will be instrumental in understanding the role of atomic motion and its coupling with the electronic motion in fundamental chemical processes such as energy transfer, molecular conformational changes, and chemical reactions.

## Methods

### Sample and tip preparation

The experiments were performed in a custom-built scanning tunneling microscope (STM) operating in ultra-high vacuum conditions (~$2 \times 10^{-10}$ mbar), and at liquid helium temperature (~10 K). Au(111) surfaces were prepared by repeated cycles of sputtering with 1.0 keV Ar$^+$ ions and thermal annealing at ~500 °C. Au tips were prepared by electrochemical etching in a concentrated HCl solution using a homemade electronic control circuit[61]. The tips were further modified by atomistic modification through tip-indentation on the Au(111) surface to achieve strong plasmonic enhancement. H$_2$Pc molecules were thermally sublimated onto the NaCl-covered Au(111) surface using a homemade evaporator with the substrate kept at liquid helium conditions (~10 K). All topographic images presented in this work were acquired in the constant current mode of the STM.

### Spectroscopic measurements

The optical setup of the STM-based near-field spectroscopic measurements is shown in Supplementary Fig. 4. A Helium-Neon (He-Ne) CW laser (HNL150L, Thorlabs) centered at ~633 nm was used for the TEPL measurements. A supercontinuum white light fiber laser (SuperK FIANIUM) was used to generate the continuously wavelength-tuned laser beam. An achromatic lens (diameter: 50 mm; focusing length: 75 mm) was mounted inside the UHV chamber to focus the laser beams onto the apex of the Au tip. The molecular Raman and fluorescence signals were collected through the same achromatic lens and then focused into the entrance slit of a spectrometer (Kymera 328i, ANDOR) and detected by a thermoelectrically cooled charge-coupled device (iDus 416, ANDOR). A dichroic mirror centered at ~650 nm was used to separate the laser beam and the molecular signal.

### Resonance Raman calculations

The electronic structure DFT calculations have been performed with the Gaussian 16 package[62] using the B3LYP functional and the 631G(d,p) basis set. The resonance Raman scattering calculations have been carried out by fully taking into account the excited electronic state involved in the transition as implemented in the Gaussian 16 package[63,64] (see the Supplementary Section II for more details). First, a geometry optimization of the HPc$^-$ molecule in the electronic ground state was carried out and the vibrational frequencies and eigenvectors of the different normal modes in this electronic ground state were obtained. Then, the vertical electronic excitation spectrum and transition dipole moments at the ground state geometry were calculated within a Time-dependent-DFT (TDDFT) approach[65,66], obtaining up to

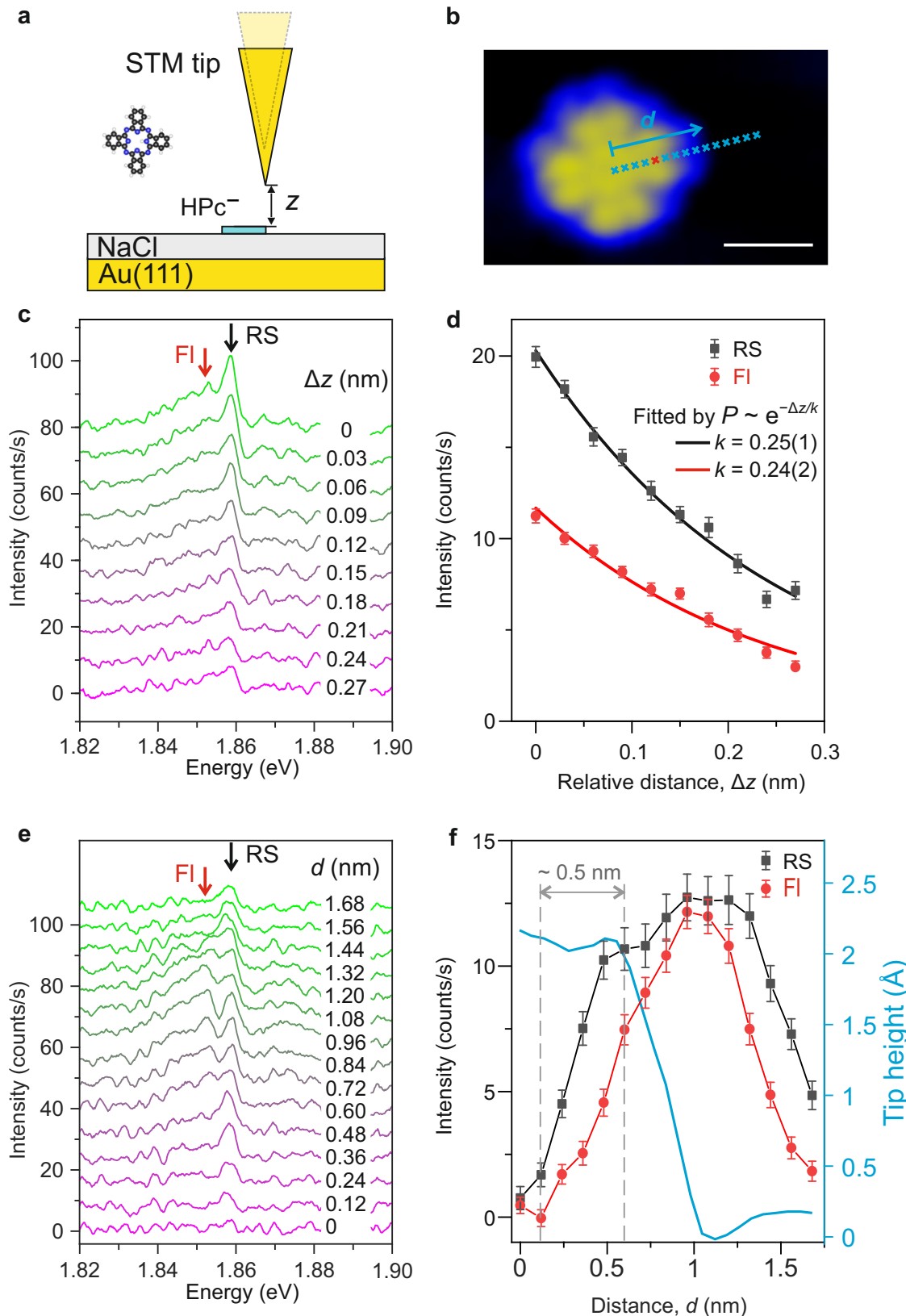

the 20[th] electronic state (see Table I in the SM). This procedure allowed us to identify the electronic transition from the $S_0$ to the $S_1$ states involved in the experiments. A geometry optimization was then performed for the molecule in the $S_1$ state, using as starting point the ground state geometry. For this configuration, a vibrational analysis was carried to obtain the vibrational energies and eigenvectors, which were plugged into the resonance Raman scattering calculation. In the Raman spectra, the energy of the vibrational modes was scaled by a factor of 0.96, determined by the comparison between the experimental Raman spectrum from $H_2Pc$ powder and the calculated Raman spectrum of the $H_2Pc$ molecule for the ground electronic state (Supplementary Fig. 6).

**Fig. 4 | Ångström-scale resolution in selective excitation of molecular vibrations.** Schematic illustration for measuring the dependence of the resonance Raman scattering (RS) and fluorescence (Fl) signals upon changing the plasmonic gap size (**a**) and the lateral position (**b**) of the nanotip over the HPc⁻ molecule. The white scale bar indicates a length of 1 nm. **c** TEPL spectra measured by retracting the nanotip with a step size of 30 pm over the molecular lobe (red cross in **b**). The STM junction was stabilized at $V = -1.5$ V, $I = 4$ pA before starting the measurement ($\Delta z = 0$ nm). **d** Variation in the spectral intensities of the resonance Raman scattering (black squares, averaged over 1.856–1.861 eV) and fluorescence signals (red dots, averaged over 1.840–1.850 eV) upon relative change of the plasmonic gap size $\Delta z$. **e** Molecular spectra recorded by placing the nanotip over the blue crosses shown in (**b**), with a step size of 1.2 Å. **f** Variation in the spectral intensities of the

resonance Raman scattering (black squares, averaged over 1.856–1.861 eV) and fluorescence signals (red dots, averaged over 1.840–1.850 eV) upon change of the lateral position of the nanotip over the molecule. The spatial resolution is estimated to be ~ 5 Å. The vertical dashed gray curves indicate the lateral distances corresponding to the 10% and 90% intensity levels of the Raman scattering signal from molecular center to the first plateau. The STM was operated in the constant current mode at $V = -1.5$ V, $I = 4$ pA. The blue curve represents the relative height profile of the nanotip during the measurement. The spectra in (**c**) and (**e**) are vertically shifted for clarity. The laser wavelength was set at 633.5 nm, the laser power was kept constant at 0.23 mW, and the acquisition time was $t = 20$ s for all the spectra. Error bars in (**d**) and (**f**) show the standard deviation of the background noise.

## Data availability
The data that support the findings of this study are provided as a Source Data file. Source data are provided with this paper.

## Code availability
The DFT, TD-DFT and Resonance Raman calculations have been carried out using the Gaussian16 package[62]. The geometries of the molecular structures used in this study are now provided in the Source data file. The details needed to reproduce the computations have been provided in the "Methods" section and Supplementary Information file.

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

## Acknowledgements

We thank Wolfgang Stiepany and Marko Memmler for technical support. We gratefully acknowledge the support of Saunak Das and Simon Krause for measuring the UV-Vis absorption spectrum of $H_2Pc$ molecules. This article is based upon work from the COST action CA18222—Attosecond Chemistry (AttoChem), supported by COST (European Cooperation in Science and Technology). All calculations were performed at the Centro de Computación Científica de la Universidad Autónoma de Madrid (CCC-UAM). M.P. acknowledges financial support by Centro Nazionale di Ricerca in High-Performance Computing, Big Data and Quantum Computing, PNRR 4 2 1.4, CI CN00000013, CUP H23C22000360005. F.M. acknowledges support by the Ministerio de Ciencia e Innovación MICINN (Spain) through projects PID2022-138288NB-C31 and the "Severo Ochoa" Programme for Centres of Excellence in R&D (CEX2020- 001039-S). A.M.-J. acknowledges funding from HORIZON-MSCA-2022-PF-01-01 under the Marie Skłodowska-Curie grant agreement No. 101108851.

## Author contributions

Y.L., S.S., M.G., A.M.J. built the experimental set-up, performed the experiments and analyzed the experimental data. M.P. and F.M. designed and performed the theoretical calculations and analyzed the theoretical data. M.G. conceived the project and designed the experiments. K.K. supervised the project. All authors interpreted the results and contributed to the preparation of the manuscript.

## Funding

## Competing interests

The authors declare no competing interests.
