## [Peer Review File · Nature Communications]

Selective excitation of vibrations in a single moleculeREVIEWER COMMENTS

Reviewer #1 (Remarks to the Author):

In the manuscript titled "Selective Excitation of Vibrations in a Single Molecule," the authors have extended the capability of single molecule level STM-based tip-enhanced Raman spectroscopy and tip-enhanced photoluminescence by coupling the STM-tip-sample junction with picosecond laser pulses to achieve mode-specific vibrational excitation in a single phthalocyanine molecule. Overall, the manuscript is well structured, and the main conclusions are convincing. My main concern is the limited novelty of this work in terms of either the studied molecular system or technical advances. This work attempted to explain the origin of the narrow-linewidth TEPL peaks using a tunable laser excitation source and finally established the peaks as TERS peaks in resonance Raman condition. This approach has already been well explored in some previous works, e.g., Jaculbia, R.B., Imada, H., Miwa, K. et al. Single-molecule resonance Raman effect in a plasmonic nanocavity. *Nat. Nanotechnol.* 15, 105–110 (2020). Consequently, the reported methodology in the present study involving using tunable laser frequencies to trigger mode-specific vibrational excitations does not convey new insights into either scientific concepts or technology applications. Given these issues, I cannot recommend publication in *Nature Communications*. I would recommend that the authors submit this to a more specialized journal, such as *ACS Nano* or *ACS Photonics*.

Other issues to be addressed include the follows:

- 1) The authors have claimed a spatial resolution of ~ 5 Ångström in the selective excitation of molecular vibrations in a single molecule. Neither in the manuscript's main text nor in the SI the authors have provided the method they used to determine the spatial resolution.
- 2) Referring to Fig. 2a the authors have mentioned that the broad fluorescence peak remains at the same energy upon changing the excitation wavelength of the laser. Here, the 'broad fluorescence peak' is not recognizable in Fig. 2a.
- 3) The information on the spectral resolution of the TEPL peaks obtained using the CW laser is missing from the manuscript. The authors mentioned, "We note that the picosecond laser pulses have a spectral linewidth of ~ 1.5 nm (~ 5 meV at 640 nm), which reduces the spectral resolution in the TEPL spectra as compared with the CW laser excitation (Fig. 1b)." For this comparison between the TEPL linewidth of CW laser and pulsed laser a quantitative measurement of the spectral resolution of TEPL peaks obtained using CW laser should be provided.
- 4) The description of fluorescence in the form "whereas the red arrow and the space between the tilted dashed-red lines indicate the position of the broad fluorescence emission" in Fig. 2b is unclear. Why is the fluorescence evolving with excitation energy?
- 5) Is 4 pA a sufficient amount of current to obtain TEPL at CW laser excitation at $\lambda \sim 633$ nm, laser power of 0.2 mW, $V = -1$ V, $I = 4$ pA, and $t = 10$ s?

Reviewer #2 (Remarks to the Author):

This is a very interesting and important study. The authors succeeded in demonstrating selective excitation of the vibrational modes by tuning the excitation wavelength of the laser to be resonant with the corresponding molecular vibronic transitions between the ground electronic state and specific vibrational modes in the excited electronic state. The originality and novelty of the reported study are high, and the manuscript is well written. Thus, I recommend the publication of this manuscript in *Nature Comm.* after some revision.

I have the following comments to improve the manuscript.

- 1) The TEPL spectrum of the H2Pc molecule shows additional distinct narrow peaks over a broad peak possibly induced by the internal Stark effect in the deprotonated molecule. Explain the last

part more clearly.

2) The authors should show UV-Vis spectra of H2Pc and HPC- in SI with brief description of their electronic transitions.

3) Why did the authors use excitation wave length in the region of 631 nm to 641.5 nm? The authors had better use a wider region.

4) The authors should explain the results in Fig. 3d more clearly. For example, what is the green dashed line in the top panel? How did the authors simulate the spectra in Fig. 3d?

5) In the explanation of Fig. 3d, the authors mentioned "differences in the relative spectral intensities of the measured (red curve) and calculated (green curve) spectra might be attributed to the effect of the nanocavity plasmons on the molecular vibronic transition rates." The authors need to explain this conclusion with stronger evidence.

6) The authors need more detailed description of the fabrication and characteristics of the gold tip used. Why did they use the gold tip not silver tip?

7) Conclusion is rather too simple. It should contain the impact of this study on a wide area of chemistry.

Response to the referees

We would like to cordially thank both the reviewers for their invaluable comments, which helped us improve the manuscript significantly. We have revised the manuscript in accordance to their suggestions and criticisms. We mark essential revisions as **REV#** and have highlighted modifications in the main-text and supplementary materials to ease the evaluation.

We hope that the reviewers will acknowledge the improvements made in the manuscript and will now be happy to recommend it for publication in Nature Communications. Below we enlist detailed responses to all comments of the reviewers.

Reviewer #1 (Remarks to the Author):

In the manuscript titled “Selective Excitation of Vibrations in a Single Molecule”; the authors have extended the capability of single molecule level STM-based tip-enhanced Raman spectroscopy and tip-enhanced photoluminescence by coupling the STM-tip-sample junction with picosecond laser pulses to achieve mode-specific vibrational excitation in a single phthalocyanine molecule. Overall, the manuscript is well structured, and the main conclusions are convincing. My main concern is the limited novelty of this work in terms of either the studied molecular system or technical advances. This work attempted to explain the origin of the narrow-linewidth TEPL peaks using a tunable laser excitation source and finally established the peaks as TERS peaks in resonance Raman condition. This approach has already been well explored in some previous works, e.g., Jaculbia, R.B., Imada, H., Miwa, K. et al. Single-molecule resonance Raman effect in a plasmonic nanocavity. Nat. Nanotechnol. 15, 105-110 (2020). Consequently, the reported methodology in the present study involving using tunable laser frequencies to trigger mode-specific vibrational excitations does not convey new insights into either scientific concepts or technology applications. Given these issues, I cannot recommend publication in Nature Communications. I would recommend that the authors submit this to a more specialized journal, such as ACS Nano or ACS Photonics.

We would like to thank the reviewer for his/her careful review and for acknowledging that our conclusions are convincing. We totally agree with the reviewer that the resonance Raman effect is a well-established methodology for enhancing signal levels of Raman scattering in the bulk systems, and Jaculbia et al. [Ref. 21] demonstrated the resonance Raman effect at the single-molecule level, when **the excitation photon energy matches the molecular optical gap**. In single-molecule studies, laser excitation with photon energies higher than the optical gap are frequently used, **as in the present work**, to generate Raman scattering [Refs. 25, 26], molecular luminescence [Refs. 43, 45], and to induce structural changes [Ref. 47]. Therefore, a fundamental question that remains to be answered is how a molecule reacts to laser excitation under this condition. **Does a molecule absorb a photon followed by intramolecular transitions (e.g. vibrational relaxation followed by electronic transition)? OR does it scatter the photon, resulting in excitation of a vibrational state? How to distinguish the Raman transitions from purely electronic transitions? Which of these processes dominate the light-molecule interaction?**

In our work, we have demonstrated state-selective enhancement of the resonance Raman signals from single molecules through mode-specific vibronic excitations, **which has not been explored elsewhere**. We distinguished the contributions of the Raman scattering and fluorescence processes by their emission peak positions and linewidths in a single molecule, which, to the best of our knowledge, has not been reported until date. **Our approach allows precisely controlling the specific vibrational modes being excited in a single molecule, thus providing insights into the light-matter interactions at the single-molecule level.** Therefore, we believe our work advances the understanding of single-molecule science and may enable future applications in mode-selective enhancement of chemical reactions and enable a better understanding of energy transfer dynamics.

Other issues to be addressed include the follows:

1) The authors have claimed a spatial resolution of ~ 5 Ångström in the selective excitation of molecular vibrations in a single molecule. Neither in the manuscript's main text nor in the SI the authors have provided the method they used to determine the spatial resolution.

We sincerely thank the reviewer for pointing this out. The spatial resolution is estimated from the variation of the Raman scattering signal when making a lateral line scan over the molecule (please see Fig. 4b (main-text)). The Raman scattering intensity profile shows a minimum at the molecular center, and reaches a plateau at around 0.6 nm where the nanotip height is almost constant. At larger distances from the molecular center (> 1 nm), the Raman scattering intensity further increases due to the nanotip approaching closer to the Au (111) surface. We have determined the spatial resolution in the selective excitation of molecular vibrations by estimating the separation between the spatial points in the lateral line scan where the Raman signal level changes from 10% to 90% from molecular center to the first plateau.

In the revised Fig. 4f, we have now indicated the lateral distances corresponding to the 10% and 90% intensity levels of the Raman signal. Please see **REV#1** in the caption of Fig. 4 on page 20.

Fig. R1: The variation in the spectral intensities of the resonance Raman scattering and fluorescence signals upon change of the lateral position of the nanotip over the molecule. The

spatial resolution is estimated to be $\sim 5 \text{ \AA}$. The vertical dashed grey curves indicate the lateral distances corresponding to the 10% and 90% intensity levels of the Raman scattering signal from molecular center (0.12 nm) to the first intensity plateau (0.6 nm).

2) Referring to Fig. 2a the authors have mentioned that the broad fluorescence peak remains at the same energy upon changing the excitation wavelength of the laser. Here, the 'broad fluorescence peak'; is not recognizable in Fig. 2a.

We would like to thank the reviewer for this comment. The Raman spectra in Fig. 2a demonstrate the distinct evolution of the Raman peaks as a function of the excitation wavelength. The relatively weaker and broader fluorescence contribution is less visible in this plot. However, in the surface plot of the Raman excitation map (Fig. 2b); the broad fluorescence contribution is clearly visible, as indicated by the red arrow.

In the revised manuscript, we have modified the discussion related to Fig. 2a. Please see **REV#2** in the main-text on page 5.

“An apparent gradual shift of the narrow emission peaks, as indicated by the colored arrows in Fig. 2a, can be clearly seen upon decreasing the excitation wavelength of the laser.”

3) The information on the spectral resolution of the TEPL peaks obtained using the CW laser is missing from the manuscript. The authors mentioned, “We note that the picosecond laser pulses have a spectral linewidth of $\sim 1.5 \text{ nm}$ ($\sim 5 \text{ meV}$ at 640 nm), which reduces the spectral resolution in the TEPL spectra as compared with the CW laser excitation (Fig. 1b.” For this comparison between the TEPL linewidth of CW laser and pulsed laser a quantitative measurement of the spectral resolution of TEPL peaks obtained using CW laser should be provided.

The spectral resolution of the TEPL spectra measured with CW laser excitation is estimated to be $\sim 15 \text{ cm}^{-1}$ ($\sim 2 \text{ meV}$), from the FWHM of the Raman peaks (**Fig. R2**). This information is now included in the revised Supplementary Fig. 5. Please see **REV#3** in the supplementary materials.

Fig. R2: Comparison between the measured TEPL spectrum from an HPc^- molecule (red curve) with CW laser and the calculated Raman spectrum for the ground electronic state (blue curve). The spectral resolution of TEPL spectra measured with CW laser excitation is estimated to be $\sim 15 \text{ cm}^{-1}$ ($\sim 2 \text{ meV}$), from the FWHM of the Raman peak (indicated by the green double arrows).

4) The description of fluorescence in the form “whereas the red arrow and the space between the tilted dashed-red lines indicate the position of the broad fluorescence emission” in Fig. 2b is unclear. Why is the fluorescence evolving with excitation energy?

Using the spectral axis (x-axis) representing light emission wavelength, the fluorescence peak remains spectrally invariable; however, the Raman peaks change their spectral position on the variation of the excitation photon energy. By converting the horizontal axis in Fig. 2a from photon energy (eV) to relative Raman shift frequency (cm^{-1}) with respect to the laser excitation wavelength, we produced a Raman excitation map shown in Fig. 2b. In this map, the otherwise constant broad fluorescence peak appears to spectrally shift with the excitation wavelength when viewed in the representation of Raman shift spectral axis. Nevertheless, the narrow linewidth Raman peaks appear spectrally invariable, i.e. they maintain their constant spectral shift with respect to the excitation photon energy.

To avoid any confusion for the readers, we have modified the discussion on the fluorescence peak. Please see **REV#4** in the main-text on page 5.

“The broad fluorescence peak, which is spectrally invariable in the photon energy representation, appears to gradually shift with the excitation wavelength in the Raman shift representation.”

5) Is 4 pA a sufficient amount of current to obtain TEPL at CW laser excitation at $\lambda \sim 633 \text{ nm}$, laser power of 0.2 mW, $V = -1 \text{ V}$, $I = 4 \text{ pA}$, and $t = 10 \text{ s}$?

Yes, TEPL signal levels from electronically decoupled molecules in an STM are usually quite strong. Other groups (Ref. 43 and Ref. 45) have used similar experimental conditions for the single-molecule TEPL measurements.

Reviewer #2 (Remarks to the Author):

This is a very interesting and important study. The authors succeeded in demonstrating selective excitation of the vibrational modes by tuning the excitation wavelength of the laser to be resonant with the corresponding molecular vibronic transitions between the ground electronic state and specific vibrational modes in the excited electronic state. The originality and novelty of the reported study are high, and the manuscript is well written. Thus, I recommend the publication of this manuscript in Nature Comm. after some revision.

We would like to sincerely thank the reviewer for his/her appreciation of our work, highly constructive suggestions, and for recognizing the novelty of our work.

I have the following comments to improve the manuscript.

1) The TEPL spectrum of the H₂Pc molecule shows additional distinct narrow peaks over a broad peak possibly induced by the internal Stark effect in the deprotonated molecule. Explain the last part more clearly.

The TEPL and STML spectra of the HPc⁻ molecule show a fluorescence peak at ~1.86 eV, which is blue-shifted with respect to the fluorescence peak of the H₂Pc molecule (~1.815 eV). This blue shift of the fluorescence emission is attributed to the Stark effect due to the electrostatic field generated by the internal charges of the deprotonated HPc⁻ molecule. This phenomenon has been studied in detail in an earlier work (*Nature Communications* **13**, 677 (2022) [Ref. 54]).

We have modified the discussion on this apparent spectral shift. Please see **REV#5** in the main-text on page 4.

“The STML spectrum of the HPc⁻ molecule shows a blue-shifted fluorescence peak at ~1.86 eV (blue curve in the bottom panel of Fig. 1b) compared to the H₂Pc molecule, plausibly induced by the internal Stark effect owing to the electrostatic field generated by the internal charges of the deprotonated HPc⁻ molecule⁵⁴.”

2) The authors should show UV-Vis spectra of H₂Pc and HPc⁻ in SI with brief description of their electronic transitions.

We would like to thank the reviewer for this suggestion. We now show the UV-Vis absorption spectrum of H₂Pc molecules dissolved in the chlorobenzene solvent (Fig. R3), and have added the spectrum and the relevant discussion in the supplementary materials (Please see **REV#6**). The single HPc⁻ molecules were obtained by deprotonation of H₂Pc molecules in the STM. We cannot produce HPc⁻ molecules in a solvent to measure its UV-Vis spectrum. We hope for the understanding of the reviewer.

Fig. R3: UV-Vis absorption spectrum of H₂Pc molecules dissolved in Chlorobenzene solvent. The concentration of the molecules in the solvent was ~ 0.1 mg/ml. The UV-Vis spectrum of H₂Pc molecules shows two strong absorption regions, one in the UV region at about 300–400 nm (B band) and the other in the visible region at about 600–700 nm (Q band), including the characteristic splitting of the Q band into Q_x and Q_y transitions. The peak spectral positions of the Q_x and Q_y bands are red-shifted in the UV-Vis spectrum with respect to the single molecule TEPL spectrum due to the effect of the solvent.

3) Why did the authors use excitation wavelength in the region of 631 nm to 641.5 nm? The authors had better use a wider region.

Extending the TEPL measurements to a broader excitation wavelength region of the laser would alter the beam alignment (including the beam focusing), necessitating further adjustment of the laser spot in the STM junction during one measurement run. Additionally, the vibrational features around 1200 cm⁻¹ are too close to each other, exceeding the available spectral resolution of our tunable ps laser source (~ 40 cm⁻¹). Therefore, we have focused on investigating the vibrational features between 500 and 900 cm⁻¹, which are rather discrete with significant spectral separation between them, using the laser excitation wavelength in the range from 630 to 641.5 nm.

4) The authors should explain the results in Fig. 3d more clearly. For example, what is the green dashed line in the top panel? How did the authors simulate the spectra in Fig. 3d?

We thank the reviewer for pointing out this issue. We have modified the relevant paragraph in the main-text to explain the results in Fig. 3d more clearly. Please see **REV#7** in the main-text on page 7.

“Five distinct Raman modes can be distinguished by fitting the experimental data (black curve) with five narrow Gaussian peaks (dashed-red curves) on a broad Gaussian background (dashed-gray curve). The experimentally measured vibrational features can be well-reproduced in the simulated Raman spectrum (green curve) obtained by summing up all the TEPL spectra in the simulated Raman excitation map (Fig. 3c), especially for the frequencies of the vibrational modes.”

5) In the explanation of Fig. 3d, the authors mentioned “differences in the relative spectral intensities of the measured (red curve) and calculated (green curve) spectra might be attributed to the effect of the nanocavity plasmons on the molecular vibronic transition rates.” The authors need to explain this conclusion with stronger evidence.

We thank the reviewer for this insightful comment. The theoretical calculations assume a uniform field and neglect the spectral distribution of the field. Whereas in the experiment, the molecules are subjected to a highly localized electric field generated by the nanocavity plasmons, which can strongly affect the molecular spectral features, as discussed in some recent works: *Nature* **568**, 78–82 (2019); *J. Phys. Chem. C* **126**, 12121–12128 (2022). Therefore, the relative peaks intensities in the experimental and calculated spectra can be different.

Now we have modified the discussion on how plasmonic enhancement can affect the spectral features. Please see **REV#8** in the main-text on page 7.

“In the experiment, the molecules are subjected to a highly localized plasmonic field in the STM junction, while in the DFT calculations the field is assumed to be spatially uniform and with no spectral distribution. Both spatial and spectral distributions of the field^{26,27} and the local geometry of the molecule⁶⁰ can affect the measured spectral features, which may contribute to the differences in the relative spectral intensities between the measured and calculated spectra.”

6) The authors need more detailed description of the fabrication and characteristics of the gold tip used. Why did they use the gold tip not silver tip?

Spectral overlap of the nanocavity plasmonic resonance with the molecular vibronic transitions is crucial for measuring TERS, STML, and TEPL spectra from a single molecule. Typically, silver nanotips produce strong plasmonic resonance in the range from 500 to 600 nm, while gold tips can generate nanocavity plasmons at longer wavelengths. Therefore, we have used gold nanotips for better local enhancement in the 600 to 700 nm range. However, it is worth noting that the plasmonic resonance of silver nanotips can also be modified to this spectral range through atomistic modifications.

We have now included more detailed description of the nanotip preparation. Please see **REV#9** in the supplementary materials.

“Au tips were prepared by electrochemical etching in a concentrated HCl solution using a homemade electronic control circuit⁶¹. The tips were further treated by atomistic modification through tip-indentation on the Au (111) surface to achieve strong plasmonic enhancement.”

7) Conclusion is rather too simple. It should contain the impact of this study on a wide area of chemistry.

Thank you very much for this suggestion. We have now extended the conclusion section to emphasize the impact of our work. Please see **REV#10** in the main-text on page 8.

“This methodology provides a unique avenue for probing the photoexcitation processes in single molecules, offering insights into the intricate interactions between light and matter at the sub-molecular level. Moreover, our ability to control the population of selective vibrational modes will be instrumental in understanding the role of atomic motion and its coupling with the electronic motion in fundamental chemical processes such as energy transfer, molecular conformational changes, and chemical reactions.”

REVIEWERS' COMMENTS

Reviewer #1 (Remarks to the Author):

The authors have addressed my comments regarding the methodology, data analysis and interpretation. I don't have further questions.

Reviewer #2 (Remarks to the Author):

The authors replied to my comments and suggestions nicely. The revised version is significantly better than the original one. So, now I can recommend the publication of this version in NC.